# Synthesizing Informative Training Samples with GAN

**Bo Zhao, Hakan Bilen**
School of Informatics, The University of Edinburgh
`{bo.zhao, hbilen}@ed.ac.uk`

## Abstract

Remarkable progress has been achieved in synthesizing photo-realistic images with generative adversarial networks (GANs). Recently, GANs are utilized as the training sample generator when obtaining or storing real training data is expensive even infeasible. However, traditional GANs generated images are not as informative as the real training samples when being used to train deep neural networks. In this paper, we propose a novel method to synthesize Informative Training samples with GAN (IT-GAN). Specifically, we freeze a pre-trained GAN model and learn the informative latent vectors that correspond to informative training samples. The synthesized images are required to preserve information for training deep neural networks rather than visual reality or fidelity. Experiments verify that the deep neural networks can learn faster and achieve better performance when being trained with our IT-GAN generated images[1]. We also show that our method is a promising solution to dataset condensation problem.

## 1 Introduction

In the last decade, generative adversarial networks (GANs) (9; 28; 35; 67; 5; 17) have been successfully applied to synthesize photo-realistic images in various tasks, including for generating novel realistic images (16; 5), image manipulation (50; 12), image-to-image translation (14; 67), text-to-image translation (37; 60), super-resolution (22; 56) and photo inpainting (33; 25). The main focus of these works has been on improving the reality and fidelity of GAN generated images. More recently, the interest of the community has shifted into turning GANs into infinite training data generators. To this end, GANs have been used to synthesize labelled training samples for part segmentation (61; 57; 24; 23), forming memory for previously seen tasks in continual learning (41; 53; 7), distilling/transferring knowledge (8; 26; 51), augmenting existing real data (2; 4; 42; 40) and reducing privacy leakage (55; 34). Nevertheless, the common assumption in these works, on which little attention has been paid before (36), is that GAN synthesized images are inherently informative for training models.

In this work, we question this assumption and ask whether the objective of generating images that are expected to be real-looking (*i.e.* by using a discriminator loss) is sufficient to train deep networks from scratch. We hypothesize that generating realistic images does not automatically guarantee good training samples, and we propose a GAN based method, *IT-GAN* that can generate more *I*nformative *T*raining samples such that a model trained on them yields better generalization performance (illustrated in Fig. 1). In particular, we first show that training a standard convolutional network (*i.e.* ResNet18 (11)) from scratch on a state-of-the-art GAN (*i.e.* BigGAN (5)) synthesized images performs significantly worse than training them on the original real training images (*i.e.* 77.8% v.s. 93.4% testing accuracy on CIFAR10 (20)). We also study an alternative strategy and show that learning latent vectors to reconstruct the original real images by GAN inversion (1; 66; 54) improves the informativeness of synthetic images for training deep models, while still performing significantly

---

[1]The implementation is available at `https://github.com/VICO-UoE/IT-GAN`.

NeurIPS 2022 Workshop on Synthetic Data for Empowering ML Research.

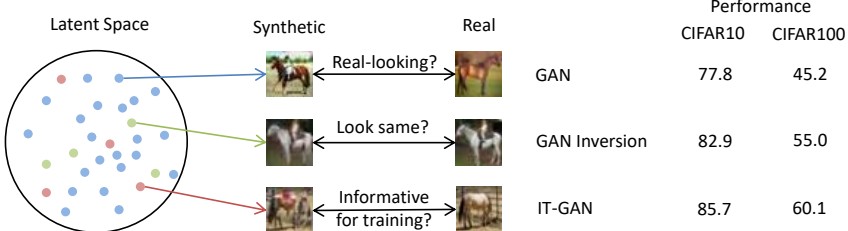

Figure 1: Different objectives and performances of traditional GAN, GAN inversion (1) and our IT-GAN. BigGAN (5) architecture is used in all three methods. We learn latent vectors in the latent space of a pre-trained GAN that correspond to informative training samples. We show that our IT-GAN achieves better performance than traditional GAN and GAN inversion on CIFAR10/100 when being used as the training data generator.

worse than the training on real images. Motivated by this observation, we propose to learn a set of latent vectors, which are fed into a pre-trained GAN to generate informative training images, such that their representations are statistically similar to those of real images with the same embedding model. In contrast to generating realistic images, our method ensures that the synthesized images include similar discriminative patterns to the original training images which in return enables more effective training of downstream task models.

Our method is most related to the recently emerging problem of dataset distillation/condensation (49; 64) that aims to synthesize a small number of informative training samples so that deep neural networks trained on synthetic samples can obtain comparable generalization performance to those trained on real samples. Most of dataset condensation methods (49; 46; 3; 64; 62; 31; 32; 48; 6) involves the expensive bilevel optimization, in which the outer-loop and inner-loop optimize the synthetic data and neural network parameters respectively. Recently, (63) propose to match the distribution of real and synthetic data in many randomly sampled embedding spaces, thus it involves neither bi-level optimization nor second-order derivative. This method significantly reduces the synthesis cost while achieving comparable performance.

Inspired by (63), we learn to generate informative training samples with our IT-GAN by minimizing the distribution matching loss of real and generated synthetic samples. Different from (63) that optimizes image pixels directly, our method optimizes latent vectors of a pre-trained GAN, so that our method can convert any pre-trained GAN into an informative training sample generator. In experiments, we verify that our method can generate more informative training samples than the traditional GAN and GAN inversion, so that deep neural networks can learn faster on our synthetic images and achieve better testing performance on CIFAR10 and CIFAR100 datasets. We also compare IT-GAN to dataset condensation method (63) and show that our IT-GAN achieves better performance with same storage budget. In the remainder of the paper, we present our method in Sec. 2, evaluate our method in multiple image classification benchmarks in Sec. 3 and conclude the paper in Sec. 4. The appendix presents the preliminary of GANs and dataset condensation, and also the ablation study.

## 2 Method

### 2.1 Initializing with GAN Inversion

Given a pre-trained generator $G$, we initialize the whole latent set $\mathcal{Z} \in \mathbb{R}^{|\mathcal{T}| \times d_z}$ by GAN inversion, so that the synthetic image $G(z)$ of each latent vector $z \in \mathbb{R}^{d_z}$ corresponds to the real image $x \in \mathbb{R}^{d_I}$ in the training set $\mathcal{T} = \{x_i, y_i\}|_{i=1}^{|\mathcal{T}|}$, where $d_z$ and $d_I$ are the dimensions of latent vector and image respectively. We use the GAN inversion method proposed in (1) which learns the latent vectors by minimizing both feature and pixel distances between the synthetic image and real image:

$$\arg\min_{z} \frac{1}{d_f} \|\psi_{\boldsymbol{\vartheta}}(G(z)) - \psi_{\boldsymbol{\vartheta}}(x)\|^2 + \frac{\lambda_{pixel}}{d_I} \|G(z) - x\|^2, \qquad (1)$$

where $\psi_{\boldsymbol{\vartheta}}$ is a pre-trained feature extractor, $d_f$ is the feature dimension and $\lambda_{pixel} = 1$ by default.

### 2.2 Condensing Training Information

Motivated by dataset condensation methods, we condense the training knowledge from real images into synthetic images that are generated by $G$. Furthermore, we learn the optimal latent vector set by

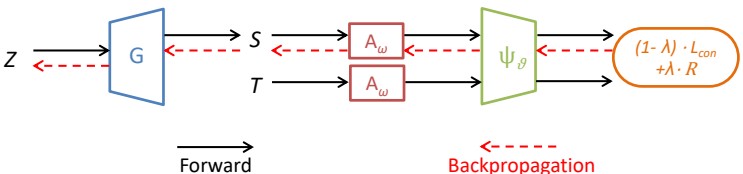

Figure 2: Illustration of IT-GAN. We input the latent vector set $\mathcal{Z}$ into a pre-trained generator $G$ and generate the synthetic set $\mathcal{S}$. Then, the synthetic set $\mathcal{S}$ and real training set $\mathcal{T}$ are input into the differentiable Siamese augmentation $\mathcal{A}_\omega(\cdot)$ and then randomly sampled embedding function $\psi_\vartheta(\cdot)$ to obtain feature embeddings, where $\omega$ and $\vartheta$ are the parameters. The condensation loss $\mathcal{L}_{con}$ and regularization $R$ with coefficient $\lambda$ are computed for optimizing $\mathcal{Z}$.

minimizing the condensation loss (illustrated in Fig. 2). As the large latent vector set $\mathcal{Z}$ has the same or comparable scale (instance number) as the whole training set $\mathcal{T}$, the condensation optimization has to be simple and fast. Thus, we leverage the condensation method proposed in (63) which has neither bi-level optimization nor second-order derivation. Specifically, the synthetic samples are expected to have similar distribution to that of real training samples in randomly sampled embedding space $\psi_\vartheta$:

$$\mathcal{L}_{con} = \mathbb{E}_{\substack{\vartheta \sim P_\vartheta \\ \omega \sim \Omega}} \| \frac{1}{|\mathcal{T}|} \sum_{i=1}^{|\mathcal{T}|} \psi_\vartheta(\mathcal{A}(\boldsymbol{x}_i, \omega)) - \frac{1}{|\mathcal{Z}|} \sum_{j=1}^{|\mathcal{Z}|} \psi_\vartheta(\mathcal{A}(G(\boldsymbol{z}_j), \omega)) \|^2, \tag{2}$$

where the differentiable augmentation $\mathcal{A}_\omega$ parameterized with $\omega \sim \Omega$ is applied to increase the data-efficiency (62). Similarly, we can also match the mean gradients $\frac{1}{|\mathcal{T}|} \sum_{i=1}^{|\mathcal{T}|} g_\vartheta(\mathcal{A}(\boldsymbol{x}_i, \omega), y_i)$ and $\frac{1}{|\mathcal{Z}|} \sum_{j=1}^{|\mathcal{Z}|} g_\vartheta(\mathcal{A}(G(\boldsymbol{z}_j), \omega), y_j)$ of real and synthetic samples (64), where $y_i$ is the label and $g$ is gradient function. We empirically verify that distribution matching and gradient matching have close performances.

## 2.3 Regularization

As the whole latent vector set size $|\mathcal{Z}|$ is large, we need to split $\mathcal{Z}$ into many batches $B_i^{\mathcal{Z}}$ and train each batch independently. Training multiple batches (or subsets) with the same condensation loss $\mathcal{L}_{con}$ will enforce the different batches to be homogeneous and thus decrease the informativeness when combining them for training. To avoid this problem, we further add the regularization:

$$R = \mathbb{E}_{\substack{\vartheta \sim P_\vartheta \\ \omega \sim \Omega}} \| \| \psi_\vartheta(\mathcal{A}(\boldsymbol{x}_i, \omega)) - \psi_\vartheta(\mathcal{A}(G(\boldsymbol{z}_i), \omega)) \|^2, \tag{3}$$

where $\boldsymbol{x}_i$ and $\boldsymbol{z}_i$ are a pair. Different from the feature alignment in GAN inversion methods, our regularization can better preserve the training information as it is calculated over the randomly sampled embedding spaces and Siamese augmentation strategies which can mimic the training dynamics. The total training loss is

$$\mathcal{L} = (1 - \lambda) \cdot \mathcal{L}_{con} + \lambda \cdot R, \tag{4}$$

where $\lambda$ is the coefficient of regularization.

## 2.4 Training Algorithm

The training algorithm is illustrated in Alg. 1. Given the pre-trained generator $G$, we freeze its parameters. We initialize a set of learnable latent vectors $\mathcal{Z}$ by implementing GAN inversion using Eq. 1. Then, we sample a batch of latent vectors $B_c^{\mathcal{Z}} \sim \mathcal{Z}$ and corresponding real image batch $B_c^{\mathcal{T}} \sim \mathcal{T}$ for each class $c$. We also sample an independent large-batch $\tilde{B}_c^{\mathcal{T}} \sim \mathcal{T}$ for condensing training information. The augmentation parameter $\omega_c \sim \Omega$ is sampled for all three batches. Then, we compute the condensation loss $\mathcal{L}_{con}$ and regularization $R$ respectively. The latent vector set $\mathcal{Z}$ is optimized by minimizing the total loss Eq. 4. Note that the synthetic samples $G(\boldsymbol{z})$ are generated instantaneously before computing the loss.

**Latent Vector Ensemble** Training the whole latent vector set $\mathcal{Z}$ in one device (*e.g.* GPU) can be infeasible or slow, as the sample number is the same or comparable to that of the original dataset. Thus, we split the whole latent vector set into many batches and train independently and then combine them to use. Latent vector ensemble is also important strategy for real-world learning scenarios such as continual learning, curriculum learning and distributed learning.

**Algorithm 1:** IT-GAN.

---

**Input:** Training set $\mathcal{T}$

1  **Required**: Pre-trained generator $G$, latent vector set $\mathcal{Z}$ for $C$ classes, deep neural network $\psi_{\boldsymbol{\vartheta}}$ parameterized with $\boldsymbol{\vartheta} \sim P_{\boldsymbol{\vartheta}}$, differentiable augmentation $\mathcal{A}_\omega$ parameterized with $\omega \sim \Omega$, coefficient $\lambda$, training iterations $K$, learning rate $\eta$.

2  Initialize $\mathcal{Z}$ by GAN inversion using Eq. 1 and correspond to every sample in $\mathcal{T}$.

3  **for** $k = 0, \cdots, K - 1$ **do**

4     Sample $\boldsymbol{\vartheta} \sim P_{\boldsymbol{\vartheta}}$

5     Sample batch $B_c^{\mathcal{Z}} \sim \mathcal{Z}$ and corresponding $B_c^{\mathcal{T}} \sim \mathcal{T}$, large-batch $\tilde{B}_c^{\mathcal{T}} \sim \mathcal{T}$ and $\omega_c \sim \Omega$ for every class $c$

6     Compute $\mathcal{L}_{con} = \sum_{c=0}^{C-1} \| \frac{1}{|B_c^{\mathcal{T}}|} \sum_{\boldsymbol{x} \in B_c^{\mathcal{T}}} \psi_{\boldsymbol{\vartheta}}(\mathcal{A}_{\omega_c}(\boldsymbol{x})) - \frac{1}{|\tilde{B}_c^{\mathcal{Z}}|} \sum_{(\boldsymbol{z},y) \in \tilde{B}_c^{\mathcal{Z}}} \psi_{\boldsymbol{\vartheta}}(\mathcal{A}_{\omega_c}(G(\boldsymbol{z}))) \|^2$

7     Compute $R = \frac{1}{|B_c^{\mathcal{T}}|} \sum_{\boldsymbol{x} \in B_c^{\mathcal{T}}, \boldsymbol{z} \in B_c^{\mathcal{Z}}} \| \psi_{\boldsymbol{\vartheta}}(\mathcal{A}_{\omega_c}(\boldsymbol{x})) - \psi_{\boldsymbol{\vartheta}}(\mathcal{A}_{\omega_c}(G(\boldsymbol{z}))) \|^2$  ▷ each $\boldsymbol{z}$ corresponds to $\boldsymbol{x}$

8     $\mathcal{L} = (1 - \lambda) \cdot \mathcal{L}_{con} + \lambda \cdot R$

9     Update $\mathcal{Z} \leftarrow \mathcal{Z} - \eta \nabla_{\mathcal{Z}} \mathcal{L}$

**Output:** $\mathcal{Z}$

---

|  | GAN | GAN Inversion | IT-GAN | Upper-bound |
|---|---|---|---|---|
| CIFAR10 | 77.8±0.7 | 82.9±0.6 | **85.7±0.4** | 93.4±0.2 |
| CIFAR100 | 45.2±1.0 | 55.0±0.8 | **60.1±0.2** | 74.1±0.2 |

Table 1: Performance (%) comparison among traditional GAN, GAN Inversion and our IT-GAN. The synthetic images produced by the three methods are used to train ResNet18 from scratch and then test on real testing data. The upper-bound performance is achieved by training ResNet18 on the original real training set.

## 3 Experiments

### 3.1 Experimental Settings

**Experimental Settings.** We do experiments on CIFAR10 and CIFAR100 (20) datasets. The experiments have two phases. In the first phase, we learn the informative latent vectors which correspond to those informative training samples on one architecture. In the second phase, we train randomly initialized deep neural networks on synthesized images and then test on the real testing set. Following (64; 62; 63), we use ConvNet and ResNet18 (11) in experiments. ConvNet is lightweight model with 3 convolutional blocks, and each block consists of a 128-kernel convolutional layer, instance normalization (47), ReLU activation (30) and average pooling. ResNet18 is equipped with batch normalization (13). For simplicity, we train latent vectors on ConvNet and then test on ResNet18 in most experiments. We find that the learned latent vectors and their corresponding synthetic images generalize well to unseen architectures.

**Competitors.** We compare our method to traditional *GAN*: the images are generated with the randomly sampled latent vectors and *GAN Inversion* (1): the images are generated with the optimized latent vectors which reconstruct the real images in original training set. We pre-train BigGAN models (5) using the state-of-the-art training strategy (65). Besides, we also compare to dataset condensation method (63) with a similar storage budget and verify that our method achieves better performance.

**Hyper-parameters.** We use Adam optimizer (18) with learning rate $\eta = 0.001$ for all experiments, which is validated in ablation study. We train latent vectors for 5000 iterations. We use batch size 1250 and 500 for splitting latent vectors of CIFAR10 and CIFAR100 into subsets respectively, and then learn these subsets independently in main experiments. The regularization coefficient $\lambda$ can be searched from $10^{\{-4,-3,-2,-1\}}$ roughly. For simplicity, we set it to be 0 when the training batch size is large enough. Please refer to the ablation study in appendix for more details. We pre-train hundreds of ConvNets on CIFAR10 and CIFAR100 and then used in experiments. The pre-training is not expensive as ConvNet architecture is simple and small. For training neural networks, we use SGD optimizer and train for 200 epochs. The learning rate is 0.01 in the first half epochs and then decreases to 0.001 in the second half epochs. We believe the performance of our IT-GAN can be further improved by using larger batch size, carefully tuning $\eta$ and $\lambda$, and using better performing embedding functions.

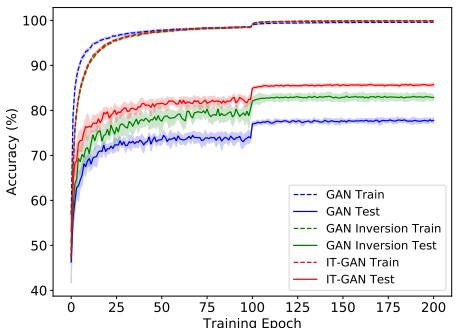
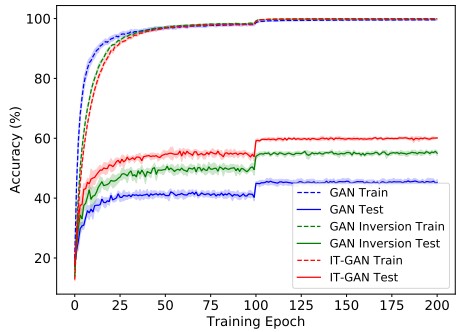

Figure 3: Train ResNet18 on synthetic CIFAR10 images produced by GAN, GAN Inversion and IT-GAN.

Figure 4: Train ResNet18 on synthetic CIFAR100 images produced by GAN, GAN Inversion and IT-GAN.

## 3.2 Comparison to GAN Methods

**Whole-set Learning.** In this setting, we sample latent vectors from the normal distribution for *GAN*. For GAN Inversion and our IT-GAN, we sample latent vectors from the whole learned latent vector set that has the same size as the real training set. The performances are presented in Tab. 1. Training ResNet18 on samples generated by traditional *GAN* achieves 77.8% and 45.2% testing accuracies on CIFAR10 and CIFAR100 respectively, while the upper-bound performances that are obtained by training on original real training set are 93.4% and 74.1% on two datasets. The significant performance gap indicates that, although the synthetic images are real-looking, they have quite different distribution from the real images which has not been revealed by the discriminator.

GAN Inversion can improve the performances by producing synthetic samples that are visually close to original ones. However, there is still big performance gap between GAN Inversion and real data training. The possible reason is that GAN Inversion tries to minimize the pixel-level difference between synthetic and real images, however some training information is lost. Our IT-GAN (85.7% and 60.1%) further improves the GAN Inversion performances (82.9% and 55.0%) by 2.8% and 5.1% on CIFAR10 and CIFAR100 respectively. It means that our method can produce more informative training samples with pre-trained GANs.

Fig. 3 and Fig. 4 plot the training and testing curves on CIFAR10 and CIFAR100 datasets. The curves show that the training accuracies of our method generated samples converge slower than the others, while the testing accuracies increase remarkably faster than the others. This training dynamics also proves that the training samples generated by our method are more informative for training models.

**Subset Learning.** To have a closer look to the informativeness of synthesized training samples and the training efficiency, we do experiments with small subsets of latent vectors. Specifically, for traditional GAN, we randomly sample a small subset of latent vectors. For GAN Inversion and our IT-GAN, we randomly learn a small subset of latent vectors. Then, the synthetic images that correspond to the sampled/learned latent vectors are used to train neural networks from scratch. Fig. 5 shows the performance curves of the three methods with varying subset size from 50 latent vectors per class to 1250 latent vectors per class. The curves verify that our IT-GAN always produces more informative training samples which have remarkable improvements over traditional GAN and GAN Inversion.

**Visualization** We visualize the synthetic images in Fig. 6. The synthetic images are recognizable, although there may exist some artificial patterns. We think those artificial patterns can improve the informativeness of training samples. Note that our goal is to generate informative training samples instead of real-looking ones.

## 3.3 Comparison to Dataset Condensation

We compare our method to dataset condensation methods under the close memory budget. Our method requires 40.8 MB storage for CIFAR10 and CIFAR100 respectively, which consists of 16.4 MB of BigGAN model and 24.4 MB of 128 dim latent vectors. This storage size is around 25% of the original dataset (162 MB). Thus, we compare to the dataset condensation methods that synthesize 25% samples for CIFAR10 and CIFAR100. However, few dataset condensation methods report the

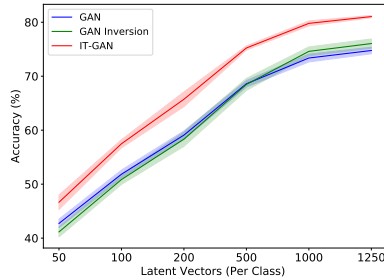

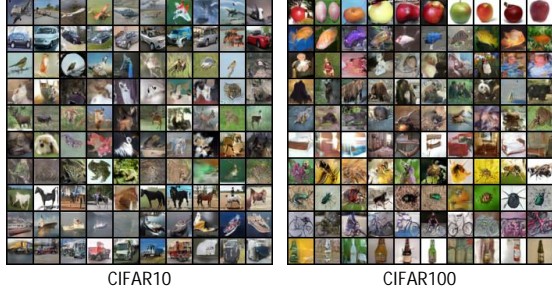

CIFAR10  CIFAR100

Figure 5: Small set of latent vectors of CIFAR10 are sampled/learned and then evaluated on ResNet18.

Figure 6: Visualization of synthetic images for CIFAR10 and CIFAR100. Note that our goal is to generate informative training samples instead of real-looking ones.

|  |  | DM | IT-GAN | Upper-bound |
|---|---|---|---|---|
| CIFAR10 | ConvNet | 80.8±0.3 | **82.8±0.3** | 84.8±0.1 |
|  | ResNet18 | 85.1±0.3 | **85.7±0.4** | 93.4±0.2 |
| CIFAR100 | ConvNet | 50.5±0.3 | **55.7±0.4** | 56.2±0.3 |
|  | ResNet18 | 56.7±0.5 | **60.1±0.2** | 74.1±0.2 |

Table 2: Compare to dataset condensation method DM (63) with the same storage (25% of the whole dataset).

results with such large synthetic set sizes due to the expensive optimization. As shown in Tab. 2, we compare to DM (63) which is simple and effective without involving bi-level optimization and second-order derivation. The results indicate that our method outperforms DM on both CIFAR10 and CIFAR100 no matter whether ConvNet or ResNet18 models are trained. Note that the synthetic data are all learned with ConvNet. Especially, on the challenging CIFAR100 dataset, our IT-GAN achieves 55.7±0.4% and 60.1±0.2% for training the two models, which exceed DM (50.5±0.3% and 56.7±0.5%) by 5.2% and 3.4% respectively. Our method can easily further reduce the storage size by reducing the latent vector dimension. Furthermore, our method is more scalable as increasing latent vectors is cheaper than increasing synthetic images especially for high-resolution images. Hence, IT-GAN is a promising solution to dataset condensation problem.

## 3.4 Cross-architecture Generalization

The learned latent vectors and their corresponding synthetic images are generic to unseen architectures. We test them on popular deep neural networks including ConvNet, VGG19 (43), ResNet18 (11), WRN-16-8 (58) and MobileNetV2 (39). The results in Tab. 3 verify that the synthetic training images work well in training all kinds of networks in downstream tasks.

|  | ConvNet | VGG19 | ResNet18 | WRN-16-8 | MobileNetV2 |
|---|---|---|---|---|---|
| CIFAR10 | 82.8±0.3 | 86.0±0.3 | 85.7±0.4 | 84.6±0.6 | 84.6±0.6 |
| CIFAR100 | 55.7±0.4 | 60.4±0.6 | 60.1±0.2 | 57.6±0.5 | 59.5±0.6 |

Table 3: Cross-architecture generalization performance (%). We learn the latent vectors with ConvNet as the feature embedding function and then evaluate the generated training images on various unseen architectures.

## 4  Conclusion

In this paper, we investigate the informativeness of GANs synthesized images for training deep neural networks from scratch. We propose IT-GAN that converts a pre-trained GAN into an informative training sample generator. Condensation loss and diversity regularization are designed to learn the informative latent vectors. Experiments on popular image datasets verify that the deep neural networks can learn faster and achieve better performance when being trained with IT-GAN generated images. We also show that our method is a promising solution to dataset condensation problem.

**Acknowledgment.** This work is funded by China Scholarship Council 201806010331 and the EPSRC programme grant Visual AI EP/T028572/1.

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

# A  Preliminary

## A.1  Generative Adversarial Networks

GANs (9) aim to synthesize photo-realistic images, which typically consist of a generator $G$ and discriminator $D$. During training, the generator and discriminator are optimized for the minimax loss function:

$$\min_G \max_D \mathbb{E}_{\boldsymbol{x} \sim P(\boldsymbol{x})}[\log D(\boldsymbol{x})] + \mathbb{E}_{\boldsymbol{z} \sim P(\boldsymbol{z})}[\log(1 - D(G(\boldsymbol{z})))], \tag{5}$$

where $P(\boldsymbol{z})$ is the distribution of latent vector $\boldsymbol{z}$ and $P(\boldsymbol{x})$ is the real data distribution. A good generator is the one that can generate images to fool the discriminator. (28) proposes a conditional GAN model that conditions each generated image on a semantic category $y$:

$$\min_G \max_D \mathbb{E}_{\boldsymbol{x} \sim P(\boldsymbol{x})}[\log D(\boldsymbol{x}|y)] + \mathbb{E}_{\boldsymbol{z} \sim P(\boldsymbol{z})}[\log(1 - D(G(\boldsymbol{z}|y)))]. \tag{6}$$

In this paper, we focus on conditional image generation task. In addition to its various applications (50; 12; 67; 60; 56; 25), the recent advances in GANs focus on increasing the training stability (38; 10; 29) and generating more diverse and real-looking images (5; 17; 59).

While one can naively employ a state-of-the-art GAN model to synthesize images for specific classes and then build a synthetic image dataset, we argue and demonstrate that the synthesized sample, despite its real-looking appearance, is not informative to train accurate deep neural networks. In other words, models trained on such synthetic data obtain significantly lower performance when being applied to real images at test time (illustrated in Fig. 1). This may be due to at least two reasons. First there can be a domain gap between synthesized and real training images, though a low discrimination loss has been achieved. Hence, the model trained on synthetic images has inferior performance on real testing images. Second the synthesized images, though optimized to look realistic, may be not as informative as real images for training purposes due to the loss of information about training. A potential way to address both issues is to find the latent vector for each synthesized image to obtain similar visual appearance to a real corresponding train image, which is investigated in GAN inversion (54).

**GAN Inversion.**  GAN Inversion (1; 66; 54) aims to find the latent vector in the the latent space of the pre-trained GAN model, which can faithfully recover a given image. Many GAN inversion methods have been proposed, and they can be roughly categorized into 3 families: optimization based inversion, learning based inversion and hybrid methods. Usually, better performance is achieved by optimization based inversion methods, as they learn latent vector for each image independently. With a pre-trained generator $G$, the latent vector for every real image is optimized by:

$$\boldsymbol{z}^* = \arg\min_{\boldsymbol{z}} \ell(G(\boldsymbol{z}) - \boldsymbol{x}), \tag{7}$$

and $\ell(\cdot)$ is the feature or pixel distance measurement. Please refer to (54) for more details. GAN inversion methods focus on the manipulation of visual effects of a specific image. The learned synthetic images are not guaranteed to have more information for training deep neural networks.

**Auto-encoder.**  Auto-encoders (19; 21; 27; 15; 44) can also generate real-looking images by learning to reconstruct real images. We believe that our method can also work well with auto-encoders, while integration with GANs may have better performances due to the better latent space of GANs. Thus, we leave the integration with auto-encoder for the future work.

## A.2  Dataset Condensation

Given a large training set $\mathcal{T} = \{\boldsymbol{x}_i, y_i\}|_{i=1}^{|\mathcal{T}|}$, dataset condensation aims to learn a small synthetic set $\mathcal{S} = \{\boldsymbol{s}_i, y_i\}|_{i=1}^{|\mathcal{S}|}$ so that the model $\boldsymbol{\theta}^{\mathcal{S}}$ trained on $\mathcal{S}$ has close generalization performance to the model $\boldsymbol{\theta}^{\mathcal{T}}$ trained on $\mathcal{T}$. Following the notations in (63), the objective is formulated as

$$\mathbb{E}_{\boldsymbol{x} \sim P_{\mathcal{D}}}[\ell(\phi_{\boldsymbol{\theta}^{\mathcal{T}}}(\boldsymbol{x}), y)] \simeq \mathbb{E}_{\boldsymbol{x} \sim P_{\mathcal{D}}}[\ell(\phi_{\boldsymbol{\theta}^{\mathcal{S}}}(\boldsymbol{x}), y)], \tag{8}$$

where the loss $\ell$ is computed on the samples from the real data distribution $P_{\mathcal{D}}$, and $\phi$ is a deep neural network parameterized with $\boldsymbol{\theta}^{\mathcal{S}}$ or $\boldsymbol{\theta}^{\mathcal{T}}$.

| Learning rate | 0.001 | 0.01 | 0.1 | 1 |
|---|---|---|---|---|
| Distribution | **55.8±1.2** | **56.8±1.1** | 52.3±0.9 | 46.5±1.2 |
| Gradient | **56.3±1.4** | 54.9±1.2 | 50.9±0.9 | 43.4±0.8 |

Table T4: Comparison of distribution and gradient matching.

**Meta-loss Based Methods.** The existing solutions to dataset condensation can be categorized based on the objective functions. (49) propose a meta-learning based method that formulates the trained model as a function of the learnable synthetic set: $\boldsymbol{\theta}^{\mathcal{S}}(\mathcal{S})$ and then minimizes the meta-loss on the real training set:

$$\mathcal{S}^* = \arg\min_{\mathcal{S}} \mathcal{L}^{\mathcal{T}}(\boldsymbol{\theta}^{\mathcal{S}}(\mathcal{S})) \quad \text{subject to} \quad \boldsymbol{\theta}^{\mathcal{S}}(\mathcal{S}) = \arg\min_{\boldsymbol{\theta}} \mathcal{L}^{\mathcal{S}}(\boldsymbol{\theta}). \tag{9}$$

This meta-learning objective has to compute the bilevel optimization and unroll the recursive computation graph. Thus, it is both time-consuming and difficult to optimize. Several methods have been proposed to improve it by introducing learnable labels (46; 3), ridge regression (3) and kernel ridge regression (31; 32).

(45) propose the generative teaching network (GTN). Specifically, they train a generator that produces informative synthetic samples and minimize the meta-loss on real training set. In experiments, they find that training generator with random or shuffled latent vectors is worse than training with deterministic sequence of latent vectors.

**Matching-loss Based Methods.** (64) propose a new framework that learns condensed synthetic data by matching the network gradients w.r.t. the real and synthetic data throughout the network optimization:

$$\mathcal{S}^* = \arg\min_{\mathcal{S}} \mathrm{E}_{\boldsymbol{\theta}_0 \sim P_{\boldsymbol{\theta}_0}} \left[ \sum_{t=0}^{T-1} D(\nabla_{\boldsymbol{\theta}} \mathcal{L}^{\mathcal{S}}(\boldsymbol{\theta}_t), \nabla_{\boldsymbol{\theta}} \mathcal{L}^{\mathcal{T}}(\boldsymbol{\theta}_t)) \right] \tag{10}$$

$$\text{subject to} \quad \boldsymbol{\theta}_{t+1} \leftarrow \texttt{opt-alg}_{\boldsymbol{\theta}}(\mathcal{L}^{\mathcal{S}}(\boldsymbol{\theta}_t), \varsigma_{\boldsymbol{\theta}}, \eta_{\boldsymbol{\theta}}),$$

where $D(\cdot, \cdot)$ computes the matching loss, $P_{\boldsymbol{\theta}_0}$ is the distribution of parameter initialization, and $T$, $\varsigma_{\boldsymbol{\theta}}, \eta_{\boldsymbol{\theta}}$ are the hyper-parameters. The new framework avoids to unroll the recursive computation graph, although it also involves bilevel optimization and second-order derivative. In addition, the synthetic data can learn from more supervision throughout the training dynamics of deep neural networks. (62) further improve the data efficiency by introducing the differentiable Siamese augmentation that enables the learned synthetic data to train deep neural networks efficiently with data augmentation. (52) learn some basic samples and combine them to form more new training samples which also improves the data efficiency. (48) design an efficient bi-level optimization algorithm with dynamic outer and inner loops. (6) propose to match training trajectories and thus mimic long-range behavior of real-data training.

Although training deep models on small synthetic sets is extremely fast, the above-mentioned bilevel optimization based condensation methods still require much computational resources to learn large synthetic sets. (63) propose a simple yet effective method without bilevel optimization and second-order derivative. Specifically, they match the distribution of real and synthetic data in many sampled embedding spaces:

$$\mathcal{S}^* = \arg\min_{\mathcal{S}} \mathbb{E}_{\substack{\boldsymbol{\vartheta} \sim P_{\boldsymbol{\vartheta}} \\ \omega \sim \Omega}} \| \frac{1}{|\mathcal{T}|} \sum_{i=1}^{|\mathcal{T}|} \psi_{\boldsymbol{\vartheta}}(\mathcal{A}(\boldsymbol{x}_i, \omega)) - \frac{1}{|\mathcal{S}|} \sum_{j=1}^{|\mathcal{S}|} \psi_{\boldsymbol{\vartheta}}(\mathcal{A}(\boldsymbol{s}_j, \omega)) \|^2, \tag{11}$$

where $\psi_{\boldsymbol{\vartheta}}$ is the embedding function parameterized with $\boldsymbol{\vartheta}$ sampled from $P_{\boldsymbol{\vartheta}}$. $\mathcal{A}(\cdot, \omega)$ is the differentiable Siamese augmentation (62) and $\omega \sim \Omega$ is the augmentation parameter. (63) also analytically connect the distribution matching with gradient matching (64). The results show that with randomly initialized neural networks as the embedding functions, the method can achieve comparable or better performance than the state-of-the-art while significantly speeding up the synthesis process.

## B  Ablation Study

We do ablation study experiments on CIFAR10 dataset. Unless otherwise stated, we train latent vectors with randomly initialized ConvNets for simplicity.

| Split | $1 \times 500$ | $2 \times 250$ | $4 \times 125$ | $5 \times 100$ |
|---|---|---|---|---|
| Accuracy | 71.8±0.8 | 71.6±0.5 | 71.1±1.0 | 70.6±0.8 |

Table T5: Performance (%) w.r.t. batch size to split latent vectors.

| $\lambda$ | 0 | 0.001 | 0.01 | 0.02 | 0.05 | 0.1 | 0.2 | 0.5 | 1 |
|---|---|---|---|---|---|---|---|---|---|
| Fixed | 70.6±0.8 | 71.1±0.5 | 71.3±0.8 | 71.3±0.7 | 70.9±0.7 | 70.4±0.7 | 70.5±0.8 | 70.0±1.0 | 69.9±0.8 |
| Random | 69.7±1.0 | 70.4±0.7 | 71.1±0.8 | 70.8±0.8 | 70.5±0.8 | 70.6±0.8 | 70.0±0.7 | 70.1±1.0 | 69.7±1.1 |

Table T6: The comparison of fixed and random splitting strategies.

**Distribution v.s. Gradient.** DC (64) and DM (63) use feature distribution and gradient to implement dataset condensation respectively. We compare the effects of using distribution and gradient as the matching objective in our method. We set regularization coefficient $\lambda = 0$ and learn 100 latent vectors per class. According to Tab. T4, the performances of distribution and gradient matching with optimal learning rate are comparable. Thus, we use distribution matching for less computational cost.

**Batch Size for Splitting Latent Vector Set.** Due to the limitation of GPU memory, we cannot load all latent vectors and corresponding synthetic images into GPU for implementing back-propagation jointly. Thus, we have to split the latent vector set and optimize the subsets. We study the relation between performance and batch size for splitting latent vectors. Given total 500 latent vectors per class, they are split into $1 \times 500$, $2 \times 250$, $4 \times 125$, $5 \times 100$ groups in four experiments. In each experiment, the different groups of latent vectors are learned independently and then combined for training neural networks. For example, in $2 \times 250$ experiment, we learn 2 independent 250 latent vectors per class sets and then combine them. $\lambda$ is set to be 0. Tab. T5 presents the results, and the results indicate that larger batch size will have better performance. The reason is that when the latent vectors are randomly split into more subsets and learned separately, they will be homogeneous in terms of the training knowledge as they are trained with the same objective.

**Fixed v.s. Mixed Latent Vector Set Splitting.** In the above ablation study, the latent vector set is split into several fixed sets and learned independently. Another possible splitting strategy is mixed splitting. It means that the latent vectors will be mixed and randomly re-split in each training iteration. In this experiment, we validate two types of splitting with fixed and random grouping respectively. We learn 500 latent vectors per class and split them into 5 subsets. Tab. T6 depicts the results of two kinds of splitting with varying $\lambda$. The results show that when the regularization coefficient $\lambda$ is small, random splitting is worse than fixed splitting. The two splitting strategies are comparable when $\lambda$ is large. This phenomenon can also be explained by the aforementioned homogenization problem, and appropriate $\lambda$ can relieve this problem by regularizing individual sample to preserve the diversity. Note that the magnitude of $\mathcal{L}_{con}$ will vary significantly for different training batch sizes, thus $\lambda$ needs to be tuned for specific training batch size.

**Regularization Coefficient.** Tab. T6 verifies that the regularization is important for learning better latent vectors. Especially, when the latent vectors are randomly grouped and optimized in each iteration, *i.e.* random splitting, each latent vector is forced to cooperate with every other one to achieve the same objective (minimizing condensation loss), which causes the homogenization problem of learned latent vectors. The regularization can largely relieve this problem.

**Network Parameter Distribution.** We can train latent vectors with feature embedding functions $\psi_{\vartheta}(\cdot)$ from different distributions $P_{\vartheta}$. We study the influence of the network parameter distribution on the learned latent vectors in Tab. T7. Following (63), we train hundreds of ConvNets and group them (including intermediate snapshots) based on their validation performances. For example, we group networks with validation performance between 50% and 60%. Then, we learn 100 latent

| Random | 10-20 | 20-30 | 30-40 | 40-50 | 50-60 | 60-70 | $\geq$70 | All |
|---|---|---|---|---|---|---|---|---|
| 55.8±1.2 | 55.8±1.4 | 56.3±0.6 | 56.4±0.5 | 57.2±0.7 | 57.0±1.0 | 57.3±0.8 | 57.3±1.1 | 57.3±1.2 |

Table T7: Learning with different network parameter distributions. Networks are grouped based on validation performances (%).

vectors per class on these network groups separately. Tab. T7 shows that our method works well on all groups of networks including randomly initialized ones. Generally speaking, networks with higher performances lead to better learned latent vectors. Specifically, training latent vectors with ConvNets that have $> 70\%$ validation accuracies achieves 57.3% testing accuracy which outperforms the result (55.8%) achieved by training with randomly initialized ConvNets by 1.5%.

