# OpenReview forum: "Synthesizing Informative Training Samples with GAN"
_NeurIPS.cc/2022/Workshop/SyntheticData4ML — Neurips 2022 SyntheticData4ML_

### Official Review · Reviewer_a8D6 · 2022-10-18
**A work that proposes to learn a set of informative GAN latent vectors**

**Rating:** 6
**Confidence:** 3

**Review:**

This work proposes a method to reduce the number of training samples used to train a downstream model. Given a pre-trained generator, the method (IT-GAN) learns an optimal subset of GAN latent vectors which minimizes the condensation loss from [63]. The experiments show that the proposed method outperforms using a standard GAN generation and GAN inversion when training the classifier, and it outperforms previous dataset condensation technique [63] on a fixed memory budget. Overall, I would consider this work borderline (leaning toward accept) - the experimental evaluation shows promise, but the method is very similar to [63] so its novelty is somewhat lacking.

Pros:
- The motivation behind the method is explained in an understandable fashion. Performing dataset condensation on latent vectors makes sense, as they take less memory than high-resolution images and as a result is more scalable (as mentioned in section 3.3).
- The experimental evaluations show that the method outperforms standard GAN baselines as well as previous methods. It is also good to see that the method transfers to various downstream classifier architectures.

Cons:
- The main issue is that it is difficult to differentiate the novelty in this work when compared to previous method [63]. The same optimization is performed as [63], but in this work it is performed on latent representations rather than images. Is this the only difference between the two methods? More discussion should be added to differentiate this work with previous works.
- In Table 3, there should be some baseline performance (e.g. upper-bound with using the entire dataset, or simply the GAN baseline), so that it is known if the method actually improves performance for these classifiers.

---

### Official Review · Reviewer_d4Qi · 2022-10-18
**This paper presents a method to create data for training purposes.**

**Rating:** 7
**Confidence:** 5

**Review:**

In this paper, the authors present IT-GAN to generate informative of GANs synthesized data for the purpose of deep learning training.
IT-GAN converts a pre-trained GAN into training sample generator. Condensation loss and diversity regularization are used to learn
the informative latent vectors. Furthermore, they run experiments on some image datasets to show that synthesizing images for training using this technique will result in better performance.
The paper largely follows ` Dataset condensation with differentiable siamese augmentation` however since their method seems useful , I think it should be accepted

---

### Official Review · Reviewer_p9jH · 2022-10-18
**Good paper with promising empirical results**

**Rating:** 8
**Confidence:** 3

**Review:**

**Paper summary**

This paper notices that traditional GANs generated images are not as informative as the real training samples when being used to train deep neural networks. To mitigate this issue, the authors propose IT-GAN, to synthesize informative training samples with GAN. The idea gives some insights for researchers to focus more on the area of dataset condensation.

**Strengths**

The motivation is clear. The paper focuses on resource management in the FL settings to improve training efficiency and resource utilization. Each part in the system design and execution section gives examples of how to achieve its design goals, which are usability, flexibility, compatibility, efficiency and scalability.

The paper gives a comprehensive description of the details of IT-GAN.

The evaluation section looks promising. The authors compare their method to traditional GAN and GAN Inversion. The experimental results show the deep neural networks can learn faster and achieve better performance when being trained with IT-GAN generated images.

The paper is well-written and easy to read.

**Weaknesses**

No significant weaknesses.

---

### Official Review · Reviewer_UJUx · 2022-10-18
**IT-GANs: Promising approach for efficiently generating synthetic data to train neural networks with smaller datasets with comparable performance to those trained with larger real datasets.**

**Rating:** 7
**Confidence:** 3

**Review:**

The authors present a method to improve the quality of synthetic GAN-generated images with respect to their information content with application towards creating smaller datasets for training downstream neural networks. The main contribution of this paper is to optimize the latent vectors of a pre-trained GAN with the goal of increasing the information-content of generated images to improve training of downstream neural networks.

Given the length constraints of the workshop, the paper is information dense and the authors provide details about their experiments in the appendix as well.

The authors also perform ablation studies showing the performance of their models and other alternative methods under different conditions and random seeds.

However, it is not mentioned how many datapoints the summary statistics have been derived from. What is the total number of networks trained under each condition? There is only a mention in line 478 of qualitative "hundreds of networks" for one specific experiment.

Also, defining the "storage budget" would be helpful to readers. The term is not present in ref 63.

Other minor issues:
- Line 2: GANs "have been" instead of "are"
- Line 8: "corresponds" instead of "correspond"
- Line 18: super-resolution "and" photo inpainting
- Line 18: "The main"
- Fig 2: Typo "Backpropagation"
- Fig 5: Typo "Latent" in x axis label
- Line 179: "involving" instead of "involve"
- Line 180: Rephrase sentence, or atleast change "not matter" to "no matter whether"
- Line 390: Typo "based"
- Line 397: Reference would be nice for the statement comparing latent space of GANs and Autoencoders.

The paper would be stronger with the checklist and societal impacts statement in line with NeurIPS guidelines (https://neurips.cc/Conferences/2022/PaperInformation/PaperChecklist), especially because synthetic data generation has broad effects in society in general.

Overall this is a promising line of research with applications in accelerating the training process for neural networks using informative synthetically generated datasets.

---

### Meta-Review · Area_Chair_jAFS · 2022-10-18

**Recommendation:** Accept